# Training Language GANs from Scratch

**Cyprien de Masson d'Autume**[*]  **Mihaela Rosca**[*]  **Jack Rae**  **Shakir Mohamed**
DeepMind
{cyprien,mihaelacr,jwrae,shakir}@google.com

## Abstract

Generative Adversarial Networks (GANs) enjoy great success at image genera-
tion, but have proven difficult to train in the domain of natural language. Chal-
lenges with gradient estimation, optimization instability, and mode collapse have
lead practitioners to resort to maximum likelihood pre-training, followed by small
amounts of adversarial fine-tuning. The benefits of GAN fine-tuning for language
generation are unclear, as the resulting models produce comparable or worse sam-
ples than traditional language models. We show it is in fact possible to train a
language GAN from scratch — without maximum likelihood pre-training. We
combine existing techniques such as large batch sizes, dense rewards and dis-
criminator regularization to stabilize and improve language GANs. The resulting
model, ScratchGAN, performs comparably to maximum likelihood training on
EMNLP2017 News and WikiText-103 corpora according to quality and diversity
metrics.

## 1   Introduction

Unsupervised word level text generation is a stepping stone for a plethora of applications, from
dialogue generation to machine translation and summarization [1, 2, 3, 4]. While recent innovations
such as architectural changes and leveraging big datasets are promising [5, 6, 7], the problem of
unsupervised text generation is far from being solved.

Today, language models trained using maximum likelihood are the most successful and widespread
approach to text modeling, but they are not without limitations. Since they explicitly model sequence
probabilities, language models trained by maximum likelihood are often confined to an autoregres-
sive structure, limiting applications such as one-shot language generation. Non-autoregressive max-
imum likelihood models have been proposed, but due to reduced model capacity they rely on distill-
ing autoregressive models to achieve comparable performance on machine translation tasks [8].

When combined with maximum likelihood training, autoregressive modelling can result in poor
samples due exposure bias [9]– a distributional shift between training sequences used for learning
and model data required for generation. Recently, [10] showed that sampling from state of the
art language models can lead to repetitive, degenerate output. Scheduled sampling [9] has been
proposed as a solution, but is thought to encourage sample quality by reducing sample diversity,
inducing mode collapse [11].

Generative Adversarial Networks (GANs) [12] are an alternative to models trained via maximum
likelihood. GANs do not suffer from exposure bias since the model learns to sample during training:
the learning objective is to generate samples which are indistinguishable from real data according to
a discriminator. Since GANs don't require an explicit probability model, they remove the restriction
to autoregressive architectures, allowing one shot feed-forward generation [13].

The sequential and discrete nature of text has made the application of GANs to language challenging,
with fundamental issues such as difficult gradient estimation and mode collapse yet to be addressed.

---

[*]Equal contribution.

Existing language GANs avoid these issues by pre-training models with maximum likelihood [14, 15, 16, 17, 18] and limiting the amount of adversarial fine tuning by restricting the number of fine-tuning epochs and often using a small learning rate [19, 20]. This suggests "that the best-performing GANs tend to stay close to the solution given by maximum-likelihood training" [20]. Even with adversarial fine-tuning playing a limited role, extensive evaluation has shown that existing language GANs do not improve over maximum likelihood-trained models [19, 20, 21].

We show that pure adversarial training is a viable approach for unsupervised word-level text generation by training a language GAN from scratch. We achieve this by tackling the fundamental limitations of training discrete GANs through a combination of existing techniques as well as carefully choosing the model and training regime. To the best of our knowledge we are the first to do so successfully; we thus call our model ScratchGAN. Compared to prior work on discrete language GANs which "barely achieve non-random results without supervised pre-training" [19], Scratch-GAN achieves results comparable with maximum likelihood models.

Our aim is to learn models that captures both both semantic coherence and grammatical correctness of language, and to demonstrate that these properties have been captured with the use of different evaluation metrics. BLEU and Self-BLEU [22] capture basic local consistency. The Fréchet Distance metric [19] captures global consistency and semantic information, while being less sensitive to local syntax. We use Language and Reverse Language model scores [20] across various softmax temperatures to capture the diversity-quality trade-off. We measure validation data perplexity, using the fact that ScratchGAN learns an explicit distribution over sentences. Nearest neighbor analysis in embedding and data space provide evidence that our model is not trivially overfitting, e.g. by copying sections of training text.

We make the following contributions:

- We show that GANs without any pre-training are comparable with maximum likelihood methods at unconditional text generation.
- We show that large batch sizes, dense rewards and discriminator regularization are key ingredients of training language GANs from scratch.
- We perform an extensive evaluation of the quality and diversity of our model. In doing so, we show that no current evaluation metric is able to capture all the desired properties of language.

The ScratchGAN code can be found at `https://github.com/deepmind/deepmind-research/scratchgan`.

## 2 Generative Models of Text

The generative model practitioner has two choices to make: how to model the unknown data distribution $p^*(\mathbf{x})$ and how to learn the parameters $\boldsymbol{\theta}$ of the model. The choice of model is where often prior information about the data is encoded, either through the factorization of the distribution, or through its parametrization. The language sequence $\mathbf{x} = [x_1, ..., x_T]$ naturally lends itself to autoregressive modeling:

$$p_{\boldsymbol{\theta}}(\mathbf{x}) = \prod_{t=1}^{T} p_{\boldsymbol{\theta}}(x_t | x_1, ..., x_{t-1}) \tag{1}$$

Sampling $\hat{x}_1, ..., \hat{x}_T$ from an autoregressive model is an iterative process: each token $\hat{x}_t$ is sampled from the conditional distribution imposed by previous samples: $\hat{x}_t \sim p_{\boldsymbol{\theta}}(x_t | \hat{x}_1, ..., \hat{x}_{t-1})$. Distributions $p_{\boldsymbol{\theta}}(x_t | x_1, ..., x_{t-1})$ are Categorical distributions over the vocabulary size, and are often parametrized as recurrent neural networks [23, 24].

The specific tokenization $x_1, ..., x_T$ for a given data sequence is left to the practitioner, with character level or word level splits being the most common. Throughout this work, we use word level language modeling.

### 2.1 Maximum Likelihood

Once a choice of model is made, the question of how to *train* the model arises. The most common approach to learn model of language is using maximum likelihood estimation (MLE):

$$\arg\max_{\boldsymbol{\theta}} \mathbb{E}_{p^*(\mathbf{x})} \log p_{\theta}(\mathbf{x}) \tag{2}$$

The combination of autoregressive models and maximum likelihood learning has been very fruitful in language modeling [5, 25, 26], but it is unclear whether maximum likelihood is the optimal perceptual objective for text data [11]. In this work we will retain the use of autoregressive models and focus on the impact of the training criterion on the quality and sample diversity of generated data, by using adversarial training instead.

## 2.2  Generative Adversarial Networks

Generative adversarial networks [12] learn the data distribution $p^*(\mathbf{x})$ through a two player adversarial game between a discriminator and a generator. A discriminator $\mathcal{D}_\phi(\mathbf{x})$ is trained to distinguish between real data and samples from the generator distribution $p_\theta(\mathbf{x})$, while the generator is trained to fool the discriminator in identifying its samples as real. The original formulation proposes a min-max optimization procedure using the objective:

$$\min_{\boldsymbol{\theta}} \max_{\boldsymbol{\phi}} \mathbb{E}_{p^*(\mathbf{x})}\big[\log \mathcal{D}_\phi(\mathbf{x})\big] + \mathbb{E}_{p_\theta(\mathbf{x})}\big[\log(1 - \mathcal{D}_\phi(\mathbf{x}))\big]. \tag{3}$$

Goodfellow et al. [12] suggested using the *alternative generator loss* $\mathbb{E}_{p_\theta(\mathbf{x})}[-\log \mathcal{D}_\phi(x)]$ as it provides better gradients for the generator. Since then, multiple other losses have been proposed [27, 28, 29, 30].

Challenges of learning language GANs arise from the combination of the adversarial learning principle with the choice of an autoregressive model. Learning $p_\theta(\mathbf{x}) = \prod_{t=1}^{T} p_\theta(x_t|x_1, ..., x_{t-1})$ using equation 3 requires backpropagating through a sampling operation, forcing the language GAN practitioner to choose between high variance, unbiased estimators such as REINFORCE [31], or lower variance, but biased estimators, such as the Gumbel-Softmax trick [32, 33] and other continuous relaxations [13]. Gradient estimation issues compounded with other GAN problems such as mode collapse or training instability [27, 34] led prior work on language GANs to use maximum likelihood pre-training [14, 15, 16, 18, 35, 36]. This is the current preferred approach to train text GANs.

## 2.3  Learning Signals

To train the generator we use the REINFORCE gradient estimator [31]:

$$\nabla_\theta \mathbb{E}_{p_\theta(\mathbf{x})}[R(\mathbf{x})] = \mathbb{E}_{p_\theta(\mathbf{x})}\big[R(\mathbf{x})\nabla_\theta \log p_\theta(\mathbf{x})\big], \tag{4}$$

where $R(\mathbf{x})$ is provided by the discriminator. By analogy with reinforcement learning, we call $R(\mathbf{x})$ a *reward*. Setting $R(\mathbf{x}) = \frac{p^*(\mathbf{x})}{p_\theta(\mathbf{x})}$, recovers the MLE estimator in Eq (2) as shown by Che et al. [17]:

$$\mathbb{E}_{p_\theta(\mathbf{x})}\left[\frac{p^*(\mathbf{x})}{p_\theta(\mathbf{x})}\nabla_\theta \log p_\theta(\mathbf{x})\right] = \mathbb{E}_{p*(\mathbf{x})}\big[\nabla_\theta \log p_\theta(\mathbf{x})\big] = \nabla_\theta \mathbb{E}_{p*(\mathbf{x})} \log p_\theta(\mathbf{x}). \tag{5}$$

The gradient updates provided by the MLE estimator can be seen as a special case of the REINFORCE updates used in language GAN training. The important difference lies in the fact that for language GANs rewards are learned. Learned discriminators have been shown to be a useful measure of model quality and correlate with human evaluation [37]. We postulate that learned rewards provide a smoother signal to the generator than the classical MLE loss: the discriminator can learn to generalize and provide a meaningful signal over parts of the distribution not covered by the training data. As the training progresses and the signal from the discriminator improves, the generator also explores other parts of data space, providing a natural curriculum, whereas MLE models are only exposed to the dataset.

Adversarial training also enables the use of domain knowledge. Discriminator ensembles where each discriminator is biased to focus on specific aspects of the samples such as syntax, grammar, semantics, or local versus global structure are a promising approach [38]. The research avenues opened by learned rewards and the issues with MLE pre-training motivate our search for a language GAN which does not make use of maximum likelihood pre-training.

## 3  Training Language GANs from Scratch

To achieve the goal of training a language GAN from scratch, we tried different loss functions and architectures, various reward structures and regularization methods, ensembles, and other modifications. Most of these approaches did not succeed or did not result in any significant gains. Via this

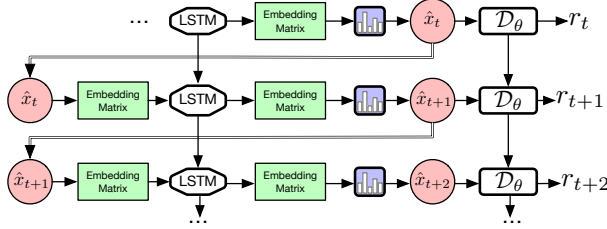

Table 1: BLEU-5 and Self-BLEU-5 metrics for a 5-gram model.

| MODEL | BLEU-5 | SBLEU-5 |
|---|---|---|
| KNESER-NEY | 20.67 | 19.73 |
| TRAINING DATA | 20.73 | 20.73 |

Figure 1: ScratchGAN architecture and reward structure.

extensive experimentation we found that the key ingredients to train language GANs from scratch are: a recurrent discriminator used to provide dense rewards at each time step, large batches for variance reduction, and discriminator regularization. We describe the generator architecture and reward structure we found effective in Figure 1 and provide a list of other techniques we tried but which proved unsuccessful or unnecessary in Appendix C.

## 3.1 Dense Rewards

Our ultimate goal is to generate entire sequences, so we could train a discriminator to distinguish between complete data sequences and complete sampled sequences, with the generator receiving a reward only after generating a full sequence. However, in this setting the generator would get no learning signal early in training, when generated sentences can easily be determined to be fake by the discriminator. We avoid this issue by instead training a recurrent discriminator which provides rewards for each generated token [35]. The discriminator $\mathcal{D}_\phi$ learns to distinguish between sentence prefixes coming from real data and sampled sentence prefixes:

$$\max_{\phi} \sum_{t=1}^{T} \mathbb{E}_{p^*(x_t|x_1,...,x_{t-1})} \left[ \log \mathcal{D}_\phi(x_t|x_1,...x_{t-1}) \right] + \sum_{t=1}^{T} \mathbb{E}_{p_\theta(x_t|x_1,...,x_{t-1})} \left[ \log(1-\mathcal{D}_\phi(x_t|x_1,...x_{t-1})) \right]$$

While a sequential discriminator is potentially harder to learn than sentence based feed-forward discriminators, it is computationally cheaper than approaches that use Monte Carlo Tree Search to score partial sentences [14, 15, 18] and has been shown to perform better empirically [19].

For a generated token $\hat{x}_t \sim p_\theta(x_t|x_{t-1}...x_1)$, the reward provided to the ScratchGAN generator at time step $t$ is:

$$r_t = 2\mathcal{D}_\phi(\hat{x}_t|x_{t-1}...x_1) - 1 \tag{6}$$

Rewards scale linearly with the probability the discriminator assigns to the current prefix pertaining to a real sentence. Bounded rewards help stabilize training.

The goal of the generator at timestep $t$ is to maximize the sum of discounted future rewards using a discount factor $\gamma$:

$$R_t = \sum_{s=t}^{T} \gamma^{s-t} r_s \tag{7}$$

Like ScratchGAN, SeqGAN-step [19] uses a recurrent discriminator to provide rewards per time step to a generator trained using policy gradient for unsupervised word level text generation. Unlike SeqGAN-step, our model is trained from scratch using only the adversarial objective, without any maximum likelihood pretraining.

## 3.2 Large Batch Sizes for Variance Reduction

The ScratchGAN generator parameters $\theta$ are updated using Monte Carlo estimates of policy gradients (Equation 4), where $N$ is the batch size:

$$\nabla_\theta = \sum_{n=1}^{N} \sum_{t=1}^{T} (R_t^n - b_t) \nabla_\theta \log p_\theta(\hat{x}_t^n|\hat{x}_{t-1}^n...\hat{x}_1^n), \qquad \hat{x}_t^n \sim p_\theta(x_t^n|\hat{x}_{t-1}^n...\hat{x}_1^n)$$

A key component of ScratchGAN is the use of large batch sizes to reduce the variance of the gradient estimation, exploiting the ability to cheaply generate experience by sampling from the generator. To

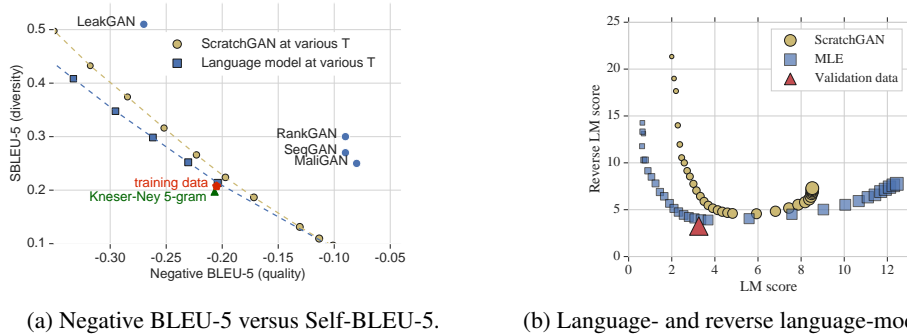

(a) Negative BLEU-5 versus Self-BLEU-5.  (b) Language- and reverse language-model scores.

Figure 2: BLEU scores on EMNLP2017 News (left) and language model scores on Wikitext-103 (right). For BLEU scores, left is better and down is better. LeakGAN, MaliGAN, RankGAN and SeqGAN results from Caccia et al. [20].

further reduce the gradient variance ScratchGAN uses a global moving-average of rewards as a baseline $b_t$ [39], as we empirically found it improves performance for certain datasets.

Providing rewards only for the sampled token as in Equation (3.2) results in a substantial training speed boost compared to methods that use $p_{\boldsymbol{\theta}}(x_t^n|\hat{x}_{t-1}^n...\hat{x}_1^n)$ to provide rewards for each token in the vocabulary, in order to reduce variance and provide a richer learning signal. These methods score all prefixes at time $t$ and thus scale linearly with vocabulary size [35].

### 3.3 Architectures and Discriminator Regularization

The ScratchGAN discriminator and generator use an embedding layer followed by one or more LSTM layers [23]. For the embedding layer, we have experimented with training the embeddings from scratch, as well as using pre-trained GloVe embeddings [40] concatenated with learned embeddings. When GloVe embeddings are used, they are shared by the discriminator and the generator, and kept fixed during training.

Discriminator regularization in the form of layer normalization [41], dropout [42] and $L_2$ weight decay provide a substantial performance boost to ScratchGAN. Our findings align with prior work which showed the importance of discriminator regularization on image GANs [34, 43, 44].

Despite using a recurrent discriminator, we also provide the discriminator with positional information by concatenating a fix sinusoidal signal to the word embeddings used in the discriminator [5]. We found this necessary to ensure the sentence length distribution obtained from generator samples matches that of the training data. Ablation experiments are provided in Appendix G.

## 4 Evaluation Metrics

Evaluating text generation remains challenging, since no single metric is able to capture all desired properties: local and global consistency, diversity and quality, as well as generalization beyond the training set. We follow Semeniuta et al. [19] and Caccia et al. [20] in the choice of metrics. We use $n$-gram based metrics to capture local consistency, Fréchet Distance to measure distances to real data in embedding space, and language model scores to measure the quality-diversity trade-off. To show our model is not trivially overfitting we look at nearest neighbors in data and embedding space.

### 4.1 $n$-gram based Metrics

BLEU [45] and Self-BLEU have been proposed [22] as measures of quality and diversity, respectively. BLEU based metrics capture local consistency and detect relatively simple problems with syntax but do not capture semantic variation [19, 46].

We highlight the limitations of BLEU metrics by training a 5-gram model with Kneser-Ney smoothing [47] on EMNLP2017-News and measuring its BLEU score. The results are reported in Table 1. The 5-gram model scores close to perfect according to BLEU-5 metric although its samples are qualitatively very poor (see Table 10 in the Appendix). In the rest of the paper we report BLEU-5

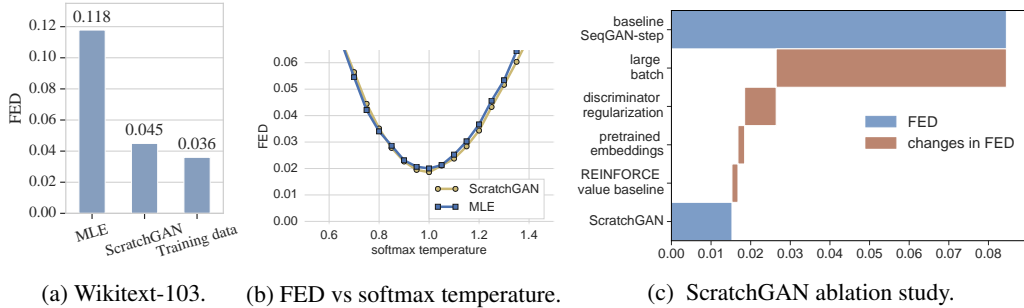

(a) Wikitext-103.　　　(b) FED vs softmax temperature.　　　(c) ScratchGAN ablation study.

Figure 3: FED scores. Lower is better. EMNLP2017 News results unless otherwise specified.

and Self-BLEU-5 metrics to compare with prior work, and complement it with metrics that capture global consistency, like Fréchet Distance.

## 4.2　Fréchet Embedding Distance

Semeniuta et al. [19] proposed the Fréchet InferSent Distance (FID), inspired by the Fréchet Inception Distance used for images [48]. The metric computes the Fréchet distance between two Gaussian distributions fitted to data embeddings, and model sample embeddings, respectively. Semeniuta et al. [19] showed that the metric is not sensitive to the choice of embedding model and use InferSent for model evaluation, while we use a Universal Sentence Encoder [49][2]. We call the metric Fréchet Embedding Distance to clarify that we use a different embedding model from Semeniuta et al. [19].

The Fréchet Embedding Distance (FED) offers several advantages over BLEU-based metrics, as highlighted in Semeniuta et al. [19]: it captures both quality and diversity; it captures global consistency; it is faster and simpler to compute than BLEU metrics; it correlates with human evaluation; it is less sensitive to word order than BLEU metrics; it is empirically proven useful for images.

We find that the Fréchet Embedding Distance provides a useful metric to optimize for during model development, and we use it to choose the best models. However, we notice that FED also has drawbacks: it can be sensitive to sentence length, and we avoid this bias by ensuring that all compared models match the sentence length distribution of the data (see details in Appendix E).

## 4.3　Language Model Scores

Caccia et al. [20] proposed evaluating the quality of generated model samples using a language model (Language Model score, LM), as well as training a language model on the generated samples and scoring the original data with it (Reverse Language Model score, RLM). LM measures sample quality: bad samples score poorly under a language model trained on real data. RLM measures sample diversity: real data scores poorly under a language model trained on samples which lack diversity. While insightful, this evaluation criteria relies on training new models, and hence the results can depend on the evaluator architecture. The metric could also have an inherent bias favoring language models, since they were trained using the same criteria.

## 5　Experimental Results

We use two datasets, EMNLP2017 News[3] and Wikitext-103 [50]. We use EMNLP2017 News to compare with prior work[15, 20] but note that this dataset has limitations: a small vocabulary (5.7k words), no out-of-vocabulary tokens, a sentence length limited to 50 tokens, and a size of only 300k sentences. Wikitext-103 is a large scale dataset of almost 4 million sentences that captures more of the statistical properties of natural language and is a standard benchmark in language modeling [51, 52]. For Wikitext-103 we use a vocabulary of 20k words. In Wikitext-103 we remove sentences with less than 7 tokens or more than 100 tokens. All our models are trained on individual sentences, using an NVIDIA P100 GPU.

| Model | World level perplexity |
|---|---|
| Random | 5725 |
| ScratchGAN | 154 |
| **MLE** | **42** |

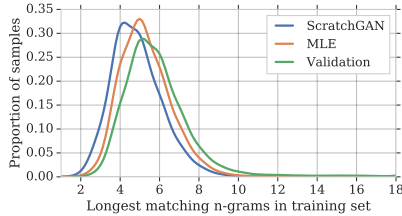

Table 2: EMNLP2017 News perplexity.    Figure 4: Matching $n$-grams in EMNLP2017.

In all our experiments, the baseline maximum likelihood trained language model is a dropout regularized LSTM. Model architectures, hyperparameters, regularization and experimental procedures for the results below are detailed in Appendix D. Samples from ScratchGAN can be seen in Appendix H, alongside data and MLE samples.

## 5.1 Quality and Diversity

As suggested in Caccia et al. [20], we measure the diversity-quality trade-off of different models by changing the softmax temperature at sampling time. Reducing the softmax temperature below 1 results in higher quality but less diverse samples, while increasing it results in samples closer and closer to random. Reducing the temperature for a language GANs is similar to the "truncation trick" used in image GANs [43]. We compute all metrics at different temperatures.

ScratchGAN shows improved local consistency compared to existing language GANs and significantly reduces the gap between language GANs and the maximum likelihood language models. Figure 2a reports negative BLEU5 versus Self-BLEU5 metrics on EMNLP2017 News for ScratchGAN and other language GANs, as reported in Caccia et al. [20].

ScratchGAN improves over an MLE trained model on WikiText-103 according to FED, as shown in Figure 3a. This suggests that ScratchGAN is more globally consistent and better captures semantic information. Figure 3b shows the quality diversity trade-off as measured by FED as the softmax temperature changes. ScratchGAN performs slightly better than the MLE model on this metric. This contrasts with the Language Model Score-Reverse Language Model scores shown in Figure 2b, which suggests that MLE samples are more diverse. Similar results on EMNLP2017 News are shown in Appendix A.

Unlike image GANs, ScratchGAN learns an explicit model of data, namely an autoregressive explicit model of language. This allows us to compute model perplexities on validation data by feeding the model ground truth at each step. We report ScratchGAN and MLE perplexities on EMNLP2017 News in Table 2. Evaluating perplexity favors the MLE model, which is trained to minimize perplexity and thus has an incentive to spread mass around the data distribution to avoid being penalized for not explaining training instances [53], unlike ScratchGAN which is penalized by the discriminator when deviating from the data manifold and thus favors quality over diversity. Improving sample diversity, together with avoiding underfitting by improving grammatical and local consistency are required in order to further decrease the perplexity of ScratchGAN to match that of MLE models.

Our diversity and quality evaluation across multiple metrics shows that compared to the MLE model, ScratchGAN trades off local consistency to achieve slightly better global consistency.

## 5.2 Nearest Neighbors

A common criticism of GAN models is that they produce realistic samples by overfitting to the training set, e.g. by copying text snippets. For a selection of ScratchGAN samples we find and present the nearest neighbors present in the training set. We consider two similarity measures, a 3-gram cosine similarity — to capture copied word sequences, and a cosine similarity from embeddings produced by the Universal Sentence Encoder —to capture semantically similar sentences. In Table 5 in Appendix B we display a selection of four random samples and the corresponding top three closest training set sentences with respect to each similarity measure, and see the training text snippets have a mild thematic correspondence but have distinct phrasing and meaning. Additionally we perform a quantitive analysis over the full set of samples; we also compare the longest matching $n$-grams between text from the training set and (a) ScratchGAN samples, (b) MLE samples, and (c) text from the validation set. In Figure 4 we see fewer ScratchGAN samples with long matching

| Table 3: FED on EMNLP2017 News. | | | Table 4: FED sensitivity on EMNLP2017 News. | |
|---|---|---|---|---|
| **Model** | **FED** | | **Variation** | **FED** |
| SeqGAN-step (no pretraining) | 0.084 | | Hyperparameters | $0.021 \pm 0.0056$ |
| ScratchGAN | **0.015** | | Seeds (best hypers) | $0.018 \pm 0.0008$ |

n-grams ($n \geq 5$) in comparison with MLE samples and text from the validation set. We conclude the generator is producing genuinely novel sentences, although they are not always grammatically or thematically consistent.

### 5.3    Ablation Study and SeqGAN-step comparison

We show the relative importance of individual features of ScratchGAN with an ablation study in Figure 3c. We successively add all elements that appear important to ScratchGAN performance, namely large batch size, discriminator regularization ($L_2$ weight decay, dropout, and layer normalization), pre-trained embeddings, and a value baseline for REINFORCE. The increase in batch size results in the most significant performance boost, due to the reduction in gradient variance and stabilizing effect on adversarial dynamics. Discriminator regularization also leads to substantial performance gains, as it ensures the discriminator is not memorizing the training data and thus is providing a smoother learning signal for the generator.

The baseline model in Figure 3c is a SeqGAN-step like model [14] without pretraining. To highlight the improvement of ScratchGAN compared to prior work, we show in Table 3 the FED difference between the two models.

### 5.4    Training Stability

Despite the high variance of REINFORCE gradients and the often unstable GAN training dynamics, our training procedure is very stable, due to the use of large batch sizes and chosen reward structure. Table 4 reports the FED scores for ScratchGAN models trained with hyperparameters from a large volume in hyper-parameter space as well as across 50 random seeds. The low variance across hyperparameters shows that ScratchGAN is not sensitive to changes in learning rate, REINFORCE discount factor, regularization or LSTM feature sizes, as long as these are kept in a reasonable range. The full hyperparameter sweep performed to obtain the variance estimates is described in Appendix F. When we fixed hyperparameters and repeated an experiment across 50 seeds, we obtained very similar FED score; no divergence or mode collapse occurred in any of the 50 runs. For WikiText-103, the results are similar ($0.055 \pm 0.003$).

## 6    Related Work

Our work expands on the prior work of discrete language GANs, which opened up the avenues to this line of research. Methods which use discrete data have proven to be more successful than methods using continuous relaxations [19], but face their own challenges, such as finding the right reward structure and reducing gradient variance. Previously proposed solutions include: receiving dense rewards via Monte Carlo Search [14, 15, 18] or a recurrent discriminator [19, 35], leaking information from the discriminator to the generator [15], using actor critic methods to reduce variance [35], using ranking or moment matching to provide a richer learning signal [16, 18] and curriculum learning [35]. Despite alleviating problems somewhat, all of the above methods require pre-training, sometimes together with teacher forcing [17] or interleaved supervised and adversarial training [15].

Nie et al. [36] recently showed that language GANs can benefit from complex architectures such as Relation Networks [54]. Their RelGAN model can achieve better than random results without supervised pre-training, but still requires pre-training to achieve results comparable to MLE models.

Press et al. [55] is perhaps the closest to our work: they train a character level GAN without pre-training. Unlike Press et al. [55], ScratchGAN is a word level model and does not require teacher helping, curriculum learning or continuous relaxations during training. Importantly, we have performed an extensive evaluation to quantify the performance of ScratchGAN, as well as measured overfitting using multiple metrics, beyond $4$-gram matching.

By learning reward signals through the use of discriminators, our work is in line with recent imitation learning work [56], as well as training non-differentiable generators [57].

# 7 Discussion

Existing language GANs use maximum likelihood pretraining to minimize adversarial training challenges, such as unstable training dynamics and high variance gradient estimation. However, they have shown little to no performance improvements over traditional language models, likely due to constraining the set of possible solutions to be close to those found by maximum likelihood. We have shown that large batch sizes, dense rewards and discriminator regularization remove the need for maximum likelihood pre-training in language GANs. To the best of our knowledge, we are the first to use *Generative Adversarial Networks to train word-level language models successfully from scratch*. Removing the need for maximum likelihood pretraining in language GANs opens up a new avenue of language modeling research, with future work exploring GANs with one-shot feedforward generators and specialized discriminators which distinguish different features of language, such as semantics and syntax, local and global structure. Borrowing from the success of GANs for image generation [43], another promising avenue is to use powerful neural network architectures [5, 54] to improve ScratchGAN.

We have measured the quality and diversity of ScratchGAN samples using BLEU metrics, Frèchet distance, and language model scores. None of these metrics is sufficient to evaluate language generation: we have shown that BLEU metrics only capture local consistency; language model scores do not capture semantic similarity; and that while embedding based Frèchet distance is a promising global consistency metric it is sensitive to sentence length. Until new ways to assess language generation are developed, current metrics need to be used together to compare models.

# 8 Acknowledgments

We would like to thank Chris Dyer, Oriol Vinyals, Karen Simonyan, Ali Eslami, David Warde-Farley, Siddhant Jayakumar and William Fedus for thoughtful discussions.

## Footnotes

[2]The model can be found at `https://tfhub.dev/google/universal-sentence-encoder/2`

[3]`http://www.statmt.org/wmt17/`

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
