[Supplementary Material]

# Supplementary material

## A   Fréchet Embedding Distance and Language model scores on EMNLP2017 News

On EMNLP2017 News, FED and LM/RLM results are similar to those on WikiText103, see Figure 5a and Figure 5b. Here we report the FED against both the training and validation set, to assess model overfitting. On this metric, we again notice that ScratchGAN performs better than the MLE model.

Figure 5: EMNLP2017 News results.

(a) FED against training and validation data.

(b) Language model scores.

## B   Nearest Neighbors

In Table 5 we see for a selection of four random samples and the corresponding top three closest training set sentences with respect to each similarity measure, there is not a clear pattern of overfitting or training set repetition.

## C   Negative results

Here we list some approaches that we tried but which proved unsuccessful or unnecessary:

- Using a Wasserstein Loss on generator logits, with a straight-through gradient. This was unsuccessful.
- Using ensembles of discriminators and generators. The results are on par with those obtained by a single discriminator-generator pair.
- Training against past versions of generators/discriminators. Same as above.
- Using bi-directional discriminators. They can work but tend to over-fit and provide less useful feedback to the generator.
- Using several discriminators with different architectures, hoping to have the simple discriminators capture simple failure modes of the generators such as repeated words. It did not improve over single discriminator-generator pair.
- Training on small datasets such as Penn Tree Bank. The discriminator quickly over-fit to the training data. This issue could probably be solved with stronger regularization but we favoured larger datasets.
- Using a Hinge loss [44] on the discriminator. This did not improve over the cross-entropy loss.
- Using a hand-designed curriculum, where the generator is first trained against a simple discriminator, and later in training a more complex discriminator is substituted. This was unsuccessful. We suspect that adversarial training requires a difficult balance between discriminator quality and generator quality, which is difficult to reach when either component has been trained independently from the other.

- Varying significantly the number of discriminator steps per generator step, say 5 discriminator steps per generator step. This was unsuccessful.

- Looking at discriminator probabilities (check that $P(real) \approx 1$ and $P(fake) \approx 0$) to evaluate training. Discriminator seems to be able to provide good gradient signal even when its predictions are not close to the targets, as long as its predictions on real data are distinct from its prediction on fake data.

- Using a population of discriminators to evaluate the quality of a generator, or conversely. This metric failed when the population as a whole is not making progress.

- Mapping all data to GloVe embeddings, and training a one-shot feed-forward generator to generate word embeddings directly, while discriminator receives word embeddings directly. This was unsuccessful.

## D   Experimental details

We now provide the experimental details of our work.

### D.1   ScratchGAN architectural details

**Generator**
The core of the generator is an LSTM with tanh activation function and skip connections. We use an embedding matrix which is the concatenation of a fixed pretrained GloVe embedding matrix of dimension $V \times 300$ where $V$ is the vocabulary size, and a learned embedding matrix of dimension $V \times M$ where $M$ depends on the dataset. An embedding for the token at the previous time-step is looked up in the embedding matrix, and then linearly projected using a learned matrix to the feature size of the LSTM. This is the input to the LSTM. The output of the LSTM is the concatenation of the hidden outputs of all layers. This output is linearly projected using a learned matrix to the dimension of the embedding matrix. We add a learned bias of dimension $V$ to obtain the logits over the vocabulary. We apply a softmax operation to the logits to obtain a Categorical distribution and sample from it to generate the token for the current time-step.

**Discriminator**
The input to the discriminator is a sequence of tokens, coming either from the real data or the generator. The core of the discriminator is an LSTM. The discriminator uses its own embedding matrix, independent from the generator. It has the same structure as the generator embedding matrix. Dropout is applied to this embedding matrix. An embedding for the token at the current time-step $t$ is looked up in the embedding matrix. A fixed position embedding of dimension $8$, depending on $t$ (see G), is concatenated to the embedding. As for the generator, the embedding is linearly projected using a learned matrix to the feature size of the LSTM. This is the input to the LSTM. The output of the LSTM is itself linearly projected to dimension $1$. This scalar is passed through a sigmoid to obtain the discriminator probability $\mathcal{D}_\phi(\mathbf{x}_t)$. The discriminator LSTM is regularized with layer normalization. $L_2$ regularization is applied to all learned variables in the discriminator.

**Losses**
The discriminator is trained with the usual cross-entropy loss. The generator is trained with a RE-INFORCE loss. The value baseline at training step $i$, denoted $b_i$, is computed as:

$$b_i = \lambda b_{i-1} + (1 - \lambda)\bar{R}_i \tag{8}$$

where $\bar{R}_i$ is the mean cumulative reward over all sequence timesteps and over the current batch at training step $i$. The generator loss at timestep $t$ and training step $i$ is then:

$$L_{ti}^G = -(R_t - b_i)\ln p_\theta(x_t) \tag{9}$$

and the total generator loss to minimize at training step $i$ is $\sum_t L_{ti}^G$.

**Optimization**
Both generators and discriminators are trained with Adam [58], with $\beta_1 = 0.5$ for both. We perform one discriminator step per generator step.

**Data considerations**
The maximum sequence length for EMNLP2017 News is 50 timesteps. The generator vocabulary also contains a special end of sequence token. If the generator outputs the end of sequence token

at any timestep the rest of the sequence is padded with spaces. At timestep $0$ the input to the generator LSTM is a space character. Generator and discriminator are both recurrent so time and space complexity of inference and training are linear in the sequence length.

## D.2 Sweeps and best hyperparameters

To choose our best model, we sweep over the following hyperparameters:

- Discriminator learning rate.
- Generator learning rate.
- Discount factor $\gamma$.
- The number of discriminator updates per generator update.
- The LSTM feature size of the discriminator and generator.
- The number of layers for the generator.
- Batch size.
- Dropout rate for the discriminator.
- Trainable embedding size.
- Update frequency of baseline, $\lambda$.

The best hyperparameters for EMNLP2017 News are:

- Discriminator learning rate: $9.38\,10^{-3}$.
- Generator learning rate: $9.59\,10^{-5}$
- Discount factor $\gamma$: 0.23.
- The LSTM feature size of the discriminator and generator: 512 and 512.
- The number of layers for the generator: 2.
- Batch size: 512.
- Dropout rate for the discriminator embeddings: 0.1
- Trainable embedding size: 64.
- Update frequency of baseline, $\lambda$: 0.08.

The best hyperparameters for WikiText-103 News:

- Discriminator learning rate: $2.98\,10^{-3}$
- Generator learning rate: $1.67\,10^{-4}$
- Discount factor $\gamma$: 0.79.
- The LSTM feature size of the discriminator and generator: 256 and 256.
- The number of layers for the discriminator: 1.
- Batch size: 768.
- Dropout rate for the discriminator embeddings: 0.4.
- Trainable embedding size: 16.
- Update frequency of baseline, $\lambda$: 0.23.

## D.3 Training procedure

For both datasets, we train for at least $100000$ generator training steps, saving the model every $1000$ steps, and we select the model with the best FED against the validation data. Each training run used approximately 4 Intel Skylake x86-64 CPUs at 2 GHz, 1 Nvidia Tesla V100 GPU, and 20 GB of RAM, for 1 to 5 days depending on the dataset.

## D.4 Language models

The language models we compare to are LSTMs. Interestingly, we found that smaller architectures are necessary for the LM compared to the GAN model, in order to avoid overfitting. For the maximum likelihood language models, we sweep over the size of the embedding layer, the feature size of the LSTM, and the dropout rate used for the embedding layer. We choose the model with the smallest validation perplexity.

For EMNLP2017 News, the MLE model used a LSTM feature size of 512, embedding size of 512, and embedding dropout rate of 0.2.

For WikiText-103, the MLE model used a LMST feature size of 3000, embedding size of 512, and embedding dropout rate of 0.3.

### D.5 Metrics

FED and BLEU/Self-BLEU metrics on EMNLP2017 News are always computed with 10000 samples. On WikiText-103 FED is computed with 7869 samples because this is the number of sentences in WikiText-103 validation data, after filtering outliers.

To compute the reverse language model scores at different softmax temperatures we used the same architecture as the best EMNLP2017 News. We trained a language model on 268590 model samples, and used it to score the validation data.

### D.6 Datasets

Wikitext-103 is available at `https://s3.amazonaws.com/research.metamind.io/wikitext/wikitext-103-v1.zip`. EMNLP2017News is available at `http://www.statmt.org/wmt17/` and a preprocessed version at `https://github.com/pclucas14/GansFallingShort/blob/master/real_data_experiments/data/news/`.

## E Fréchet Embedding Distance sensitivity to sentence length

We show that FED is slightly dependent on sentence length, highlighting a possible limitation of this metric. For each sentence length, we randomly select a subset of 10k sentences from EMNLP2017 News training set conditioned on this sentence length, and we measure the FED between this subset and the 10k validation set. We show the results in figure 6a. We see that there is a small dependence of FED on sentence length: FED seems to be worse for sentences that are significantly shorter or longer than the mean.

Figure 6: EMNLP2017 News results.

(a) FED vs sentence length.

(b) Providing positional information to the discriminator helps the generator capture sentence length distribution correctly.

## F Hyperparameter variance

Here we clarify the definition of the subset of hyper-parameter space that we used to show the stability of our training procedure. All runs with hyper-parameters in the ranges defined below gave good results in our experiments as shown in Table 4 in the main text.

- baseline decay ($\lambda$ in equation 8 in appendix D) is in $[0, 1]$.
- batch size is in $\{512, 768\}$
- discriminator dropout is in $\{0.1, 0.2, 0.3, 0.4, 0.5\}$
- discriminator LSTM feature size is in $\{256, 512, 1024\}$
- discriminator learning rate is in $[9.2 \, 10^{-5}, 3.7 \, 10^{-2}]$
- discriminator $L_2$ weight is in $\{0, 10^{-7}, 10^{-6}, 10^{-5}\}$
- discriminator LSTM number of layers is in $\{1, 2\}$
- number of discriminator updates per training step is in $\{1, 2\}$
- discount factor in REINFORCE is in $[0, 1]$
- generator LSTM feature size is in $\{256, 512\}$
- generator learning rate is in $[8.4 \, 10^{-5}, 3.4 \, 10^{-4}]$
- generator LSTM number of layers is in $\{1, 2\}$
- number of generator updates per training step is in $\{1, 2\}$
- dimension of trainable embeddings is in $\{16, 32, 64\}$

## G Positional information provided to the discriminator

Here we discuss the importance of providing positional information to the discriminator. In early experiments we noticed that the distribution of sentence length in the generator samples did not match the distribution of sentence length found in the real data. In theory, we would expect a discriminator based on a LSTM to be able to easily spot samples that are significantly too short or long, and to provide that signal to the generator. But in practice, the generator was biased towards avoiding short and long sentences.

We therefore provide the discriminator with explicit positional information, by concatenating a fix sinusoidal signal to the word embeddings used in the discriminator. We choose 8 periods log-linearly spaced $(T_1, \ldots, T_8)$ such that $T_1 = 2$ and $T_8$ is 4 times the maximum sentence length. For the token $x_t$ at position $t$ in the sentence, the positional information is $p_t^i = \sin\left(2\pi \frac{t}{T_i}\right)$. We concatenate this positional information to the word embedding for token $x_t$ in the discriminator before using it as input for the discriminator LSTM.

Figure 6b shows distributions of sentence length in samples of two GAN models, one with and one without this positional information. We compare these distributions against the reference distribution of sentence length in the training data. Even with positional information in the discriminator, the generator still seems slightly biased towards shorter sentences, compared to the training data. But the sentence length distribution is still a much better fit with positional information than without.

## H Samples

Training examples from both datasets can be found in Table 6. Samples from our model, the maximum likelihood trained language model and the $n$-gram model can be found in Tables 7, 9 and 10.

Table 5: EMNLP2017 News nearest neighbours to ScratchGAN samples. Similarity with respect to embedding cosine distance using the Universal Sentence Encoder, and with respect to 3-gram cosine distance. We see the GAN samples are not composed of cut-and-paste text snippets from the training set.

| USE | Nearest Neighbours | 3-gram | Nearest Neighbours |
|---|---|---|---|
| Sample: *A nice large part of Trump has to plan exactly what Pence would worth , for Trump to choose him strongly in Florida, where he can be 100 percent away.* | | | |
| 0.77 | His name , of course , is Donald Trump , the billionaire businessman who leads most national polls for the Republican nomination . | 0.13 | It ' s like the situation in Florida , where he didn ' t pay taxes on his golf course . |
| 0.75 | But to get there , Rubio believes he needs to cut significantly into Cruz ' s support in Iowa , a state dominated by social conservatives . | 0.12 | Donald Trump is spending his third straight day in Florida , where he ' s already made six campaign stops since Sunday . |
| 0.72 | On the Republican side , the Iowa poll shows Ted Cruz leading Donald Trump by four points , but Trump has a 16 - point lead in New Hampshire . | 0.10 | He has long been mentioned as a possible candidate for governor in Florida , where he has a home in Miami with his wife and four school - age children . |
| Sample: *I didn ' t know how to put him up to the floor among reporters Thursday or when he did what he said.* | | | |
| 0.69 | Speaking at a news conference on Monday , he said : " Let me make clear that this is a great professional and a great person . | 0.25 | Her explanation for saying " I didn ' t glass her , I don ' t know why I ' m getting arrested " was said out of panic , I didn ' t know how to handle the situation . |
| 0.67 | In a text message late Monday , he said he had not seen the court filing and could not comment on it . | 0.23 | I didn ' t know how to do it or who to talk to , so I had to create opportunities for myself . |
| 0.59 | " We ' re not going to stand by any agent that has deliberately done the wrong thing , " he said . | 0.23 | I didn ' t know how to face it , but as soon as I ' d got through that it was OK . |
| Sample: *Paul have got a fine since the last 24 game , and it ' s just a nine - day mark .* | | | |
| 0.50 | As he said after Monday night ' s game : " We know we have enough quality , it ' s not always the quality . | 0.21 | We ' ve been in this situation too many times , and it ' s a 60 - minute game , and it doesn ' t matter . |
| 0.50 | The 26 - year - old from Brisbane was forced to come from behind to score an impressive 6 - 7 ( 5 - 7 ), 6 - 4 , 7 - 6 ( 8 - 6 ) win . | 0.21 | There are already plenty people fighting fire with fire , and it ' s just not helping anyone or anything . |
| 0.48 | But he ' s had a very good start to this year and beat Roger to win Brisbane a couple of weeks ago . | 0.20 | We ' ve just got to move on , it ' s part of the game , and it ' s always going to happen , that kind of stuff |
| Sample: *Such changes from the discussion and social support of more people living in the EU with less generous income and faith.* | | | |
| 0.72 | The EU has promised Ankara three billion euros in aid if it does more to stop the flow of migrants headed for Europe . | 0.14 | There are nearly three - quarters of a million British people living in Spain and over two million living in the EU as a whole . |
| 0.68 | Now , as Norway is not a member of the EU , it has no say over these or any other EU rules . | 0.1 | About 60 people living in the facility were moved to another part of the building for safety , according to authorities . |
| 0.67 | We can ' t debate the UK ' s place in Europe ahead of an historic EU referendum without accurate statistics on this and other issues . | 0.1 | We ' d like to hear from people living in the country about what life as a Canadian is really like . |

Table 6: Training data examples on EMNLP2017 News and WikiText-103.

**EMNLP2017 News**

My sources have suggested that so far the company sees no reason to change its tax structures , which are perfectly legal .

I ' d say this is really the first time I ' ve had it in my career : how good I feel about my game and knowing where it ' s at .

We would open our main presents after lunch ( before the Queen ' s speech ) then take the dog for a walk .

**WikiText-103**

the actual separation of technetium @-@ N from spent nuclear fuel is a long process .

she was launched on N december N , after which fitting @-@ out work commenced .

covington was extremely intrigued by their proposal , considering eva perón to be a non @-@ commercial idea for a musical .

Table 7: Randomly selected ScratchGAN samples on EMNLP2017 News and WikiText-103.

**EMNLP2017 News**

We are pleased for the trust and it was incredible , our job quickly learn the shape and get on that way .

But I obviously have him with the guys , maybe in Melbourne , the players that weren ' t quite clear there .

There is task now that the UK will make for the society to seek secure enough government budget fund reduce the economy .

Keith is also held in 2005 and Ted ' s a successful campaign spokeswoman for students and a young brothers has took an advantage of operator .

Police said how a Democratic police officer , would choose the honor of alcohol and reduce his defense and foundation .

We do not go the Blues because that I spent in ten months and so I didn ' t have a great job in a big revolution .

The 28 - year - old - son Dr Price said she would have been invited to Britain for her " friend " in a lovely family .

And as long as it is lower about , our families are coming from a friend of a family .

**WikiText-103**

the general manager of the fa cup final was intended for the final day as a defensive drive , rather than twenty field goals .

the faces of competitive groups and visual effects were in much of the confidence of the band at *UNK* 's over close circles , and as well as changing the identical elements to the computing .

a much *UNK* ground was believed to convey *UNK* other words , which had been *UNK* writing and that he possessed receiving given powers by his *UNK* transport , rather than rendered well prior to his " collapse of the local government .

the highest viewership from the first N @.@ N % of the debate over the current event .

the housing of the county were built in the county behind the new south park at lake london , which , as thirty @-@ two @-@ lane work used for a new property .

near a time : bootleg was used by the brazilian navy and the german ¡unk¿ copper .

the next day , curry and defenses weren ' t , with the labour government waterfall , the powerful rock *UNK* , calling him heavy @-@ action , who are shy to refuse to fight while preferred desperate oppression in alkan .

the british the united states launched double special education to its N % ;

Table 8: Randomly selected ScratchGAN samples on EMNLP2017 News as training progresses.

**Beginning of training, FED=0.54**

because kicking firm transparency accommodation Tim earnings While contribution once forever diseases O spotlight furniture intervention guidelines false Republicans Asked defeated raid - who rapid Bryant felt ago oil refused deals today dance stocks Center reviews Storm residents emerging Duke blood draw chain Law expanding code few MPs stomach ¡unk¿ countries civilians

March labour leave theft afterwards coach 1990 importance issues American revealing players reports confirmed depression crackdown Green publication violence keeps 18th address defined photos experiencing implemented Center shots practical visa felt tweeted hurt Raiders lies artist 1993 reveal cake Amazon express party although equal touch Protection performance own rule Under golden routine

**During training, FED=0.034**

Cuba owners might go him because a break in a very small - defeat City drive an Commons

Germany made it by the chairman of his supporters , who are closed in Denver and 4 average -

Nine news she scored Donald Trump , appeared to present a New -

If he did , he wants a letter of the electorate that he accepted the nomination campaign for his first campaign to join passing the election .

The former complaint she said : " whatever this means certain players we cannot have the result of the current market .

**End of training, FED=0.018**

She ' s that result she believes that for Ms . Marco Rubio ' s candidate and that is still become smaller than ever .

I hadn ' t been able to move on the surface – if grow through ,' she said , given it at a time later that time .

If Iran wins business you have to win ( Iowa ) or Hillary Clinton ' s survived nothing else since then , but also of all seeks to bring unemployment .

All the storm shows is incredible , most of the kids who are telling the girls the people we ' re not turning a new study with a challenging group .

Six months before Britain were the UK leaving the EU we will benefit from the EU - it is meeting by auto , from London , so it ' s of also fierce faith Freedom .

Table 9: Randomly selected MLE model samples on EMNLP2017 News and WikiText-103.

**EMNLP2017 News**

It came out the five days of the developing player waiting to begin the final major European championship of state - owned teams in 2015 and 2015 .

" I look from my size , you know in the most part , I ' ve been fighting every day , " she says .

When you are around mid - 2006 , you play one and train with you earlier this year and the manager would make the opposition .

She said : ' I ' d like food to be now , where my baby and children deserve to be someone ' s kids .

He ' d been very good at that , but it ' s fun , the camera have been incredibly tight - with that we can be on the ball at the beginning of his debut .

**WikiText-103**

in an interview with journalist *UNK UNK* during his death , a new specimen was brought in the *UNK* museum of modern art .

after the sets of *UNK* wear *UNK* and *UNK* ' *UNK* ' *UNK* to tell him , *UNK UNK* they play *UNK UNK* with *UNK* around a *UNK* .

after he urged players to fight for what he saw as a fantastic match , the bank sustained a fractured arm and limited injury in the regular season .

the album peaked at number eight on rolling stones ' s N .

in the *UNK* sitting on the starboard N @-@ inch , a *UNK* woman looks ( *UNK UNK* ) with an eagle during the day of all singing due to her the doors being edged far through where she *UNK* , which included *UNK* , *UNK UNK* , *UNK UNK* , and *UNK* 's motifs on the bridge .

Table 10: Randomly selected samples from an 5-gram model with Kneser-Ney smoothing.

**EMNLP2017 News**

It ' s like a ' test site will boost powerful published on the question , 60 years on the fact that at moment .

The bridge opens fire Dallas - and they ' ll be best remembered as scheduled by accident and emergency units .

The study focused on everything Donald Trump was " somebody to cope with a social events that was not wearing the result of a 1 , 2017 , will be in .

It ' s going to finish me off , when a recent poll , more than the actual match to thank the British way of the seven years .

We can be sure that has been struck off by the company , is to be completed by Smith had taken a week later , you just like , what ' s going on in everyday reflects a material drone hundreds of comments .