[Reviews · NeurIPS 2019]

Reviewer 1



- the text in the intro claims that MLE language models cannot perform one-shot language generation. this statement isn't exactly true; there has been a lot of recent work on non-autoregressive generation from conditional LMs (mostly within machine translation) that could be cited as a counterpoint here (e.g., Gu et al., ICLR 2018). this "contribution" should thus be toned down. - many typos throughout (e.g., "World level perplexity" in table 2, "model ground through" in line 244) - the evaluation results leave me unsatisfied. while ScratchGAN does seem better than other GAN-based alternatives, the perplexity difference between it and the MLE model on wikitext103 is immense. The authors attempt to explain this difference in lines 245-250, but I didn't quite catch the drift of their argument (isn't it bad that ScratchGAN does not favor diversity during training?). However, looking at the generated samples from the MLE model and ScratchGAN in the appendix, it is clear that the huge perplexity difference actually corresponds to noticeable differences in grammaticality and coherence. - Why are there so few samples provided with the ScratchGAN after training? The supplementary material should have way more samples from each model so we can judge their relative quality, especially since the evaluation metrics used here are (outside of perplexity) hard to judge. - From my perspective, it is a stretch to say that ScratchGAN performs "comparably" to MLE trained models.

Reviewer 2



Originality: There's moderate novelty in the methodological contribution mentioned above. There's nice discussion of related work for language GANs but this field moves a bit fast and there are a couple of new papers that are not mentioned. Quality: Besides the methodological contribution, the paper does a really good job trying to evaluate language GANs with many metrics to measure the diversity and quality of generated sentences. The results look great and I always appreciate a good ablation study, which the authors did, and with such great graphs to visualize the additional contributions for each technique (Fig 3). Clarity: The paper is quite well written. The experimental details are provided in the supplementary materials which helps with the reproducibility. I wish there'd be a link, anonymous, to the code however. (why not?) Significance: This paper certainly explores a missing gap of how to train GANs for natural text which is an interesting direction. To me, I'm still not entirely convinced about the superiority of language GANs for text generation over language models. As far as I know, they GANs are still substantially worse than LMs (Tevet et al, 2019). In additions, can language GANs really scale to large neural nets such as using the Transformer? The balance between the discriminator and the generator power are quite delicate and it is progressively harder to tune once the discriminator and the generators are more powerful. There's hope however (for example, BigGANs for images) but this is still a somewhat uncharted territory for GANs. Can you please address the scalability? I am glad of the improvements made so far to use GANs for language, however.

Reviewer 3



[EDIT after author rebuttal]: Thank you very much for the rebuttal, it helped clarify some of the issues, especially those regarding comparison against MLE and potential overclaiming. I've raised my score accordingly, but I still think that there needs to be more solid results. In particular, while the rebuttal notes that ScratchGAN can almost match the MLE baseline, I am not sure how strong the MLE baseline itself is. Based on sample quality, I suspect that the MLE baseline itself is quite weak and does not use more modern LM approaches (e.g. regularization). Of course, I am not saying that the authors deliberately used weak baselines, but it would be helpful to compare against stronger MLE baselines too. -------------- Strengths: - Isolating the sources of contribution was nice to see, although it would also have been nice to see this on other metrics than FID. - I appreciate the negative results in Supplemental section C. - In general the paper was very well written and easy to read/understand. Weaknesses: - The main weakness is empirical---scratchGAN appreciably underperforms an MLE model in terms of LM score and reverse LM score. Further, samples from Table 7 are ungrammatical and incoherent, especially when compared to the (relatively) coherent MLE samples. - I find this statement in the supplemental section D.4 questionable: "Interestingly, we found that smaller architectures are necessary for LM compared to the GAN model, in order to avoid overfitting". This is not at all the case in my experience (e.g. Zaremba et al. 2014 train 1500-dimensional LSTMs on PTB!), which suggests that the baseline models are not properly regularized. D.4 mentions that dropout is applied to the embeddings. Are they also applied to the hidden states? - There is no comparison against existing text GANs , many of which have open source implentations. While SeqGAN is mentioned, they do not test it with the pretrained version. - Some natural ablation studies are missing: e.g. how does scratchGAN do if you *do* pretrain? This seems like a crucial baseline to have, especially the central argument against pretraining is that MLE-pretraining ultimately results in models that are not too far from the original model. Minor comments and questions : - Note that since ScratchGAN still uses pretrained embeddings, it is not truly trained from "scratch". (Though Figure 3 makes it clear that pretrained embeddings have little impact). - I think the authors risk overclaiming when they write "Existing language GANs... have shown little to no performance improvements over traditional language models", when it is clear that ScratchGAN underperforms a language model across various metrics (e.g. reverse LM).

[Author Response · NeurIPS 2019]

We thank our reviewers for their time and valuable comments.

**Motivation** We have observed in the literature and also from personal communication at recent conferences incl. ICML
and ICLR that almost all text GAN practitioners do not believe it is possible to train a GAN using REINFORCE
on language with such high dimensional action spaces (e.g. vocabulary sizes of 10,000 or 20,000 as we have done).
Instead the area publishes and promotes complex training techniques to overcome instability, using pre-training of the
discriminator and generator; or periodic teacher forcing with a tuned schedule of regularity. Furthermore in some cases
the GAN models are heavily pre-trained and only fine-tuned as a GAN with a miniscule learning rate (e.g. "Adversarial
Feature Matching for Text Generation" Zhang et al. 2017).

We feel this paper will have a significant impact, by showing that stable training can be obtained with REINFORCE.
We think this will re-focus the community from overcoming stability to benchmarking with richer data (EMNLP and
WikiText are probably too small) and scaling with larger models — hopefully to a state where one can observe a
significant difference in sample quality.

**Overstatement** The reviewers note that some of the wording in the paper suggests that we have solved GANs for text,
which we agree is not the case. We do show ScratchGAN is producing samples of similar quality to language models
for these datasets, however it is clearly a much worse generative model than the pure MLE variant and so far, the more
compute-intensive GAN training is not providing us with a much better model. We will tone down any language which
suggests ScratchGAN outperforms MLE. But we stand by the observation that sample quality and diversity appears to
be close to the MLE model, and some metrics confirm this (BLEU, FED).

**Sample quality** We certainly agree that neither the MLE or ScratchGAN are producing groundbreaking samples -
and we mostly attribute this to choice of dataset. We ran this on two LM datasets that had been benchmarked by
prior GAN work (EMNLP and WikiText). This had the benefit of comparison to prior work for objective measures
BLEU/self-BLEU (Fig 2a). However it has the downside that the samples are quite bad and make for a difficult
qualitative comparison. We think future work should focus on scaling to larger datasets, generating larger bodies of text
and using aggregate human evaluation to provide a more objective sense of sample quality.

**Reviewer 1** We agree with your point that we are dismissing non-autoregressive language models. We will add a couple
of sentences to highlight progress in feed-forward approaches.

We have addressed these typos, thank you for noting them! Also, we have increased the tables 7 and 9 from 5 samples
per (model, dataset) to 15.

**Reviewer 2** Since there is a symbiosis in ScratchGAN between the discriminator and generator — both are recurrent
models over text, we do believe a promising direction would be to scale the dataset and model (e.g. to a transformerxl)
to obtain better quality samples.

We agree MLEs are still superior to GANs, we have not solved GANs for text but we think this work is an important
data point along the path to doing so. Please see our *Overstatement* section — in short we will tone down the claims of
the paper. Re. code release, we are in the process of trying to release a simple colab script for training, such that people
can see all of the components working.

**Reviewer 3** We agree the sample quality is not very good, we partly address this in the *Sample quality* section above.
It was not clear to us that there was a qualitative difference between MLE and ScratchGAN, some MLE samples are
quite degenerative, e.g. "*after the sets of UNK wear UNK and UNK ' UNK ' UNK to tell him , UNK UNK they play*
*UNK UNK with UNK around a UNK .*" However future work should benchmark these approaches at scale, with a larger
model and use a cumulative human evaluation to assess qualitative appearance; alongside the automatic scores.

For objective measures, we do compare to existing GAN approaches (Fig 2a). However the real objective is to have
GANs considerably outperform MLE, since it is agreed this is still the best approach for text generation.

Re. overclaiming, we agree ScratchGAN does not outperform MLE and have toned down any language that appears to
make this claim (see *Overclaim* section above). We genuinely do not want to claim that ScratchGAN solves GANs for
text, just that it is possible to train a GAN to a decent level of quality (judged by objective measures) without a complex
training procedure of pre-training, teacher forcing, Gumbel Softmax with a scheduled temeprature increase etc.

Good point re. pre-trained word embeddings. We decided to keep them because there was no change in performance
within the ablation study — for this comparative run the model was truly from-scratch ;-).

Aside from language applications, we think the result of this study — that REINFORCE can be stably trained in
this challenging setting — will be of interest also to reinforcement learning practitioners that are interested in high-
dimensional action spaces; e.g. for medical treatment prediction in electronic health record time-series. Thus we ask
you to consider this core research contribution, when reconsidering your score.

[Meta-Review · NeurIPS 2019]

This paper has required quite a bit of discussion between the reviewers. The concerns were that each individual technique proposed in the paper has been tried in the past. However, their combination enabled something which has not been shown before: training a decent text GAN model without MLE pre-training. While the submission does not provide a convincing argument for switching from MLE to GAN in text generation, it is still an important paper. While some may question where the text GAN direction will ever deliver state-of-the-art language generation models, it is an active area. Many researchers over the last couple of years have tried to train GANs from scratch and failed. I can see that this paper should be a must read for anyone working on this topic. I also find the paper well and also honestly written, it does not oversell GANs and carefully compares the model to MLE training.